# Experimental Investigation of the In-Cylinder Flow of a Compression Ignition Optical Engine for Different Tangential Port Opening Areas

**Mitsuhisa Ichiyanagi** [1,*] **, Emir Yilmaz** [1] **, Kohei Hamada** [2] **, Taiga Hara** [2] **, Willyanto Anggono** [3] **and Takashi Suzuki** [1]

1 Department of Engineering and Applied Sciences, Sophia University, Tokyo 102-8554, Japan; yilmaz@sophia.ac.jp (E.Y.); suzu-tak@sophia.ac.jp (T.S.)
2 Graduate School of Science and Technology, Sophia University, Tokyo 102-8554, Japan; k-hamada-3c8@eagle.sophia.ac.jp (K.H.); t-hara-5d1@eagle.sophia.ac.jp (T.H.)
3 Mechanical Engineering Department, Petra Christian University, Surabaya 60236, Indonesia; willy@petra.ac.id
* Correspondence: ichiyanagi@sophia.ac.jp

**Abstract:** The push for decarbonization of internal combustion engines (ICEs) has spurred interest in alternative fuels, such as hydrogen and ammonia. To optimize combustion efficiency and reduce emissions, a closer look at the intake system and in-cylinder flows is crucial, especially when a hard-to-burn fuel, such as ammonia is utilized. In port fuel injection ICEs, airflow within cylinders profoundly affects combustion and emissions by influencing the air–fuel mixing phenomenon. Adjusting intake port openings is an important factor in controlling the in-cylinder airflow. In previous experiments with a transparent cylinder, tangential and helical ports demonstrated that varying the helical port's opening significantly impacts flow velocities, swirl ratios, and swirl center positions (SCPs). In this study, we used a particle image velocimetry technique to investigate how the tangential port's opening affects intake and in-cylinder flows. Flow velocities were assessed at different planes near the cylinder head, evaluating streamline maps, turbulent kinetic energy (TKE), and SCPs. Under the given experimental conditions, swirl flows were successfully generated early in the compression stroke when the tangential port opening exceeded 25%. Our findings emphasize the importance of minimizing TKE and SCP variation for successful swirl flow generation in engine cylinders equipped with both tangential and helical ports.

**Keywords:** in-cylinder flow; PIV; swirl flow; swirl center position; turbulent kinetic energy

## 1. Introduction

Carbon-free fuels, such as hydrogen and ammonia, are promising options for sustainable propulsion, particularly for the urgent need for decarbonization in the transportation industry. Previous experimental studies on hydrogen- [1] and ammonia [2,3] -fueled engines have revealed the necessity of a different approach to intake systems. Ammonia is considered to be a "hard to burn" fuel due to its high latent heat of vaporization and slow laminar burning velocity [2]. Higher intake air temperatures were found to be necessary for the combustion process to take place. This increase in the intake air temperature affects the air-to-fuel ratio, and thus the air–fuel mixing phenomenon [3]. It is also widely acknowledged that droplet atomization and achieving a homogenous air–fuel mixture before the start of combustion is crucial for high combustion efficiency. With the utilization of alternative fuels, further analysis of in-cylinder flow is vital as the density and viscosity of these alternative fuels are different to conventional internal combustion engine (ICE) fuels. In [4,5], fuel inside the combustion chamber was less susceptible to atomization due to higher momentum and longer break-up times for liquid droplets, resulting in soot production, which deposits inside the combustion chamber. It was shown that the generation of turbulence inside the cylinders improved the atomization of the fuel droplets as well as the

promotion of rapid air–fuel mixing, which resulted in better performance and a reduction in hazardous emissions. In [6], it was shown that a high swirl intensity affects engine combustion and emissions by reducing the combustion period, reducing soot generation, and $CO_2$ emissions. In [7], the researchers investigated the influence of swirl flow on heat transfer in a diesel engine cylinder and reported a 4–12% increase in heat transfer due to the increased swirl ratio at different operating conditions of the engine. In [8], it was shown that higher heat transfer was a consequence of increasing the swirl ratio, which enhanced the power performance by 5.79% and improved the fuel consumption as well. Furthermore, the influences of the following factors on swirl flows inside cylinders were investigated: the shape of the piston bowl [9–11], the shapes and configurations of intake ports [12–15], the volume flow rates of air from the intake ports [16–20], and so forth.

In a conventional engine cylinder, there are two types of intake ports, these being the helical port and the tangential port. These ports deliver intake air with different characteristics. Computational fluid dynamics (CFD) has been used to predict the flow characteristics of these intake ports [19–22]. In [19], the results indicated that flows within the helical port have higher velocities than those within the tangential port, and flows entering into the combustion chamber from the helical port generate intense local vortical structures. On the other hand, those from the tangential port induced a significant amount of momentum with no particular vortical structure. However, airflow was diverted by the cylinder walls to generate a swirling structure on the scale of the cylinder bore. Moreover, as the various ports were throttled, there was a significant interaction between the flows from these two ports, which could result in a non-monotonic variance in swirl generation. In particular, CFD simulations revealed that the variances in swirl flow were considerably more complex when the tangential port was throttled.

In experimental analyses, there are several methods to measure gas flows inside engine cylinders, such as the hot-wire anemometry [22] and laser Doppler velocimetry [23], which are known for their excellent temporal and spatial resolutions. However, since these methods measure only one point in a flow field, it is difficult to measure flow velocities in the entire engine cylinder. Therefore, the particle image velocimetry (PIV) technique has garnered attention in terms of measuring flow velocities simultaneously at multiple points inside the engine cylinders without interfering with the flow fields [12,24–37]. In [30], the PIV technique was applied during the compression stroke to a light-duty optical diesel engine. They evaluated the swirl center positions (SCPs) in the $x$–$y$ plane depending on the inclination of the rotation axis and the piston bowl's inclination angle. They reported that the piston geometry effect might be more responsible than the intake flow effect for the tilting of SCPs. However, PIV measurements took place only during the compression stroke and in conditions where measurement planes were close to each other. Thus, it is somewhat hard to grasp the effects of the $z$-axis during the piston's reciprocating motion. In [28], a swirl control vane was used inside the helical and the tangential ports to change the swirl ratio to 2.2, 3.5, and 4.5, and it was found that a single dominant swirl flow structure tilts with respect to the cylinder axis. Their study also focused only on the compression stroke, where the eccentricity of the swirl center also caused in-cylinder flow to be asymmetric, which lowered the swirl ratio. In our previous studies, we measured the flow velocity inside cylinders when changing the engine speed and the opening area of helical ports under both the motored- and the fired-engine conditions [14,15,34,35]. We found that by changing the helical port's opening areas, SCPs do not shift significantly, and airflow velocities do not change with an increase in engine speed. In addition, as the helical port's opening area was increased, the swirl ratio was reduced, which lowered the turbulent intensity. Thus, our attention shifted to the tangential port instead, which is the main focus of the present study; there being only a few studies where the tangential port's effect was investigated experimentally. In [19], the researchers developed a computational model to simulate intake flow and its effect on in-cylinder swirl and flow structures by using different throttle angles in both the tangential and helical ports. However, this study was mainly focused on

the intake stroke. Therefore, in this study the effect of tangential port opening areas, along with their influence on compression stroke and swirl formation was investigated.

The authors have focused on the utilization of ammonia as a carbon-free fuel in conventional ICEs. The aim is to develop an ICE that can be powered solely with ammonia. However, due to its chemical properties, ammonia is considered to be hard-to-burn fuel in conventional ICEs. Thus, the authors have conducted co-combustion experiments using an ammonia–gasoline fuel mixture [3]. However, the results have shown that the percentage of ammoniain the fuel mixture can only be increased up to 67% without losing significant power output. Thus, in order to improve ammonia's flame retardancy and increase its ratio in the fuel mixture, the authors shifted their focus on the intake system to investigate the formation of swirl flow and turbulence inside the cylinder. The investigation focused on the intake port opening areas, which is considered to be an effective method resulting in minimal changes to a conventional ICE. In the present study, the PIV technique was applied to a commonly arranged intake–exhaust port four-stroke engine, at three different measurement planes to investigate the influence of the tangential port's opening area on swirl flow generation. The PIV measurements took place during the intake and compression strokes, in order to obtain a broader perspective on swirl flow generation. The helical port's opening area was set to 100% throughout the experiments, whereas the tangential port's opening area was changed using several kinds of gaskets. The data obtained from the PIV experiments were used to calculate spatially averaged velocities and TKE. Three-dimensional coordinates of SCPs were calculated, where the tilting of SCPs became clear with the piston's reciprocating motion and the tangential port's opening areas. It was found that swirl flows were successfully formed when the tangential port's opening area was 25% or more. In addition, a preliminary relationship was found between the calculated TKE and swirl flow generation, where low variances of TKE and SCPs in in-cylinder flows led to the successful formation of swirl flow.

## 2. Experimental Setup and Methodology

### 2.1. Experimental Apparatus

Figure 1 illustrates the experimental apparatus based on the optical single cylinder diesel engine, and Table 1 provides the specification of the PIV system, which was similar to our previous studies [14,15,34,35].

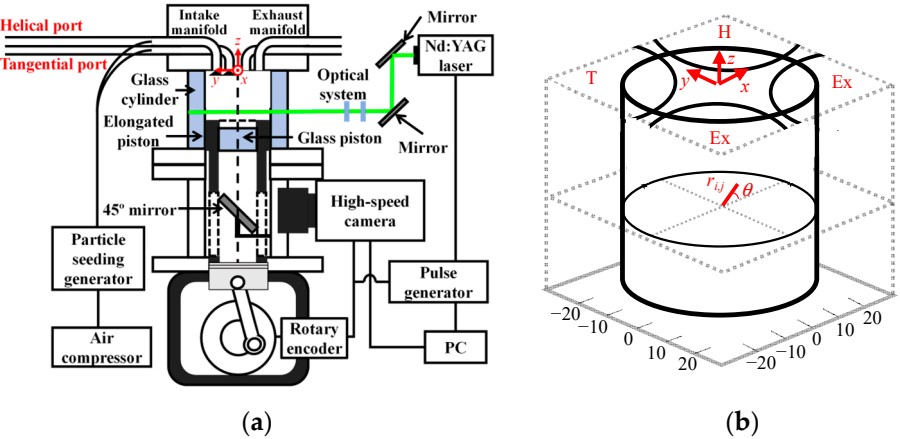

(**a**)  (**b**)

**Figure 1.** (**a**) Experimental apparatus and (**b**) location of each port on the cylinder head and its coordinates. Units are mm.

**Table 1.** Experimental engine and equipment specification.

| Equipment Name | Equipment Details | |
|---|---|---|
| Laser | Mesa-PIV (Amplitude Japan, Tokyo, Japan) Nd:YAG, Double pulse, 532 nm | |
| Particle seeding generator | PivSolid3 (PIVTEC GmbH, Göttingen, Germany) | |
| Tracer particle | Silica (SiO$_2$) particles, 4.65 μm | |
| High-speed camera | FASTCAM SA5 (Photron Ltd., Tokyo, Japan) | |
| | Spatial resolution: | 696 × 704 pixels |
| | Temporal resolution: | 15 kHz |
| Air compressor | ACP-25SLA (Takagi Co., Ltd., Niigata, Japan) | |

An optical four-valve engine with two intake ports and two exhaust ports was used during the PIV experiments. For the intake ports, the tangential port and the helical port were connected to the cylinder head of the engine as shown in Figure 2. The tangential port had a straight port which generated an airflow along the cylinder wall resulting in large-scale swirl flow. On the other hand, the helical port was highly tilted with a spiral structure before entering the cylinder, creating a small-scale swirl flow. The bore, stroke, and cavity sizes were 85 mm, 96.9 mm, and 51.6 mm, respectively.

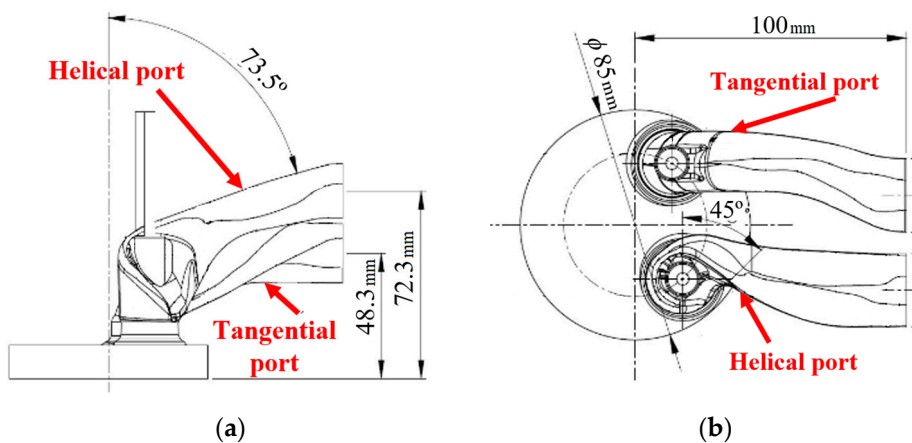

(**a**)                    (**b**)

**Figure 2.** (**a**) Side view and (**b**) top view of the helical and tangential intake ports.

For visualization inside the cylinder, the cylinder was made of transparent quartz glass and the cavity of the piston was made of sapphire glass. The PIV system had a mirror underneath the piston allowing observation inside the cylinder. Since the sapphire glass was used only in the cavity, the observable area was limited to the cavity part of the piston in the present study. From the relationship between the position of the piston and the laser light sheet, the visible crank angle (CA) was between 70 deg.CA and 290 deg.CA. For the light source, a double pulse Nd:YAG laser (Mesa-PIV, Amplitude Japan, Tokyo, Japan) was used, which radiated light with a wavelength of 532 nm. The laser light created a 1 mm thickness light sheet by passing through the cylindrical and condenser lenses. This laser light sheet was irradiated horizontally into the glass cylinder. The height of the laser light sheet was set in parallel orientation and at distances of 10 mm, 20 mm, and 30 mm below the cylinder head to capture particle images in the upper part of the engine cylinder. For the tracer particles, silica (SiO$_2$) with an average diameter of 4.65 μm was used. These particles were directed into the intake port by using a particle seeding generator (PivSolid3, PIVTEC GmbH) with an air compressor (ACP-25SLA, Takagi Co., Ltd., Niigata, Japan). When the engine was started, the tracer particles were flown into the cylinder during the intake stroke. The mirror underneath the piston reflected the scattered light inside the cylinder to the high-speed CMOS camera (FASTCAM SA5, Photron Ltd., Tokyo, Japan) to capture the tracer particle images. The high-speed CMOS camera was set at a frame rate of 15,000 frames per second and the images were captured with a spatial resolution

of 696 × 704 pixels. Images taken from this camera were analyzed using commercial software (FtrPIV, version 3.2.0.0, Flowtech Research Inc., Kanagawa, Japan) to calculate the velocity vectors of the swirl flows. In the present study, the interrogation and search window sizes were 16 × 16 pixels and 33 × 33 pixels, respectively, which overlapped by 50%. The engine, the high-speed CMOS camera, and the laser were synchronized via the pulse generator using the signal for every two deg.CA of output from the rotary encoder set in the engine. When the pulse generator received the signal from the rotary encoder once, both the laser and the camera were operated twice, which made it possible to capture of two images with a time interval of 10 μs. The error in the PIV measurements was focused on the calculation velocity of the tracer (SiO$_2$) particles. The particle flow velocity was obtained from the ratio between particles displacement, $\Delta x$, and the time interval, $\Delta t$. A measurement uncertainty at a 95% confidence interval was calculated by combining the bias and precision indices as described in [38]. For the present study, four different error factors were considered, namely $\alpha$ (magnification), $\Delta x$ (displacement of particles), $\Delta t$ (time interval), and $\delta u$ (variation of velocity based on the measurement principle). The calculation result revealed a measurement uncertainty of 1.7%, according to which the present PIV system was considered to be accurate.

### 2.2. Experimental Conditions

The experimental conditions in the present study are summarized in Table 2. For the helical port, all the gaskets had the same size of a 100% opening area. For the tangential port, five different gaskets were prepared to change the opening areas to 0%, 25%, 50%, 75%, and 100%. For the evaluation of swirl flows at different measurement planes, velocities were measured at 10 mm, 20 mm, and 30 mm below the cylinder head (which corresponds to $z = -10$ mm, $-20$ mm, and $-30$ mm, respectively). Our previous study revealed that engine speed had a minor effect on swirl flow generation [20], thus, it was set to 1000 rpm.

**Table 2.** Experimental conditions.

| | |
|---|---|
| Engine speed | 1000 rpm |
| Measurement plane | $z = -10$ mm, $-20$ mm, $-30$ mm |
| Opening area of helical port | 100% |
| Opening area of tangential port | 0%, 25%, 50%, 75%, 100% |

### 2.3. Evaluation Technique for Swirl Center Position

SCPs of flows inside engine cylinders do not stay in a fixed position but vary with each CA, which is affected by flows from the intake ports and so forth. Our successive studies developed an evaluation technique for SCPs using PIV measurements, which was based on the algorithms proposed in [29,30]. For any fixed arbitrary lattice point, $P_{i,j}$, in an in-plane velocity field, the following gamma function, $\Gamma(P_{i,j})$, was calculated, which indicates the average sine of the angle, $\theta_l$, between the vectors connecting the point to all other interrogation window centers and those to the measured velocities [37]:

$$\Gamma\left(P_{i,j}\right) = \frac{1}{N}\sum_{l=1}^{N}\sin(\theta_l) \tag{1}$$

where $N$ is the total number of lattice points in each measurement plane [29,30]. The first candidates for the SCP were determined as the lattice point with an absolute value of $\Gamma$ of more than 0.85.

The swirl ratio, $S_R$, was calculated using the PIV measurements according to the following equation, which is the ratio of the swirl flow's angular velocity to the engine's angular velocity:

$$S_R = \frac{1}{\omega} \times \frac{1}{N}\sum_{i=1}^{N}\frac{V_{i,j} \cdot \cos\theta - U_{i,j} \cdot \sin\theta}{r_{i,j}} \tag{2}$$

where $\omega$ is the engine angular velocity [rad/s], $r_{i,j}$ is the distance between the cylinder center and any lattice point [m], $\theta$ is the angle of the cylinder center and any lattice point (Figure 1b) [rad], $U_{i,j}$ and $V_{i,j}$ are the velocities in the $x$ and $y$ directions [m/s], respectively. The second candidates for the SCP were determined as the lattice point with an $S_R$ of more than 0.55. Finally, the intersections of both candidates were calculated, and the lattice point with the smallest velocity among the intersections was determined as the SCP.

### 2.4. Evaluation Technique for Turbulent Kinetic Energy

TKE, $k$, is a parameter that shows the strength of the turbulence of a flow, which is the magnitude of deviation from the average flow velocity. In the present study, TKE was calculated using PIV measurements according to the following equation:

$$k = \frac{1}{N}\sum_{l=1}^{N}\frac{\left(u'^2 + v'^2\right)}{2} \tag{3}$$

where $u'$ and $v'$ were the velocities obtained by subtracting the ensemble-averaged velocity from the instantaneous velocities in the $x$ and $y$ directions [m/s], respectively. When its deviation is large, $k$ becomes large, which means that a mixing action with a surrounding fluid was promoted. In the case of a diesel engine, a large $k$ means the promotion of the mixture of air and fuel.

### 3. Results and Discussion
### 3.1. Evaluation of the Velocity Vector and Streamline Maps

Figure 3 shows the ensemble-averaged velocity vector maps at $z = -10$ mm and $-30$ mm. Figure 4 shows the streamline maps, which were calculated using the results from Figure 3. The velocity vector calculations and streamline maps at $z = -20$ mm are omitted since the results were similar to those at $z = -10$ mm. In the present study, intake stroke was defined from 70 deg.CA to 180 deg.CA, due to the visibility constraints mentioned in the previous section. The compression stroke was divided into three periods, these being 180 deg.CA to 216 deg.CA (early period), 216 deg.CA to 254 deg.CA (middle period), and 254 deg.CA to 290 deg.CA (latter period). Table 3 shows the matching of the case numbers and port opening area. The results for 254 deg.CA were omitted since they were observed to be similar to those at 216 deg.CA. For example, the velocity vector map at 70 deg.CA under the condition of a 0% opening area was expressed as "1A". Similarly, the streamline map results at 216 deg.CA with a 75% opening area were expressed as 3I. As depicted in Figures 3 and 4, the location of the tangential port is shown as $T$, that of the helical port is shown as $H$, and that of the exhaust ports is shown as $Ex$. The origin coordinate (0, 0) indicates the cylinder center. The SCPs are plotted as white dots, which were calculated using Equation (1), and are shown in Figure 3. The analyses of in-cylinder flows were divided into intake and compression stroke cases in the next sections.

**Table 3.** Case expressions for the tangential port opening area.

| deg.CA | Case No | | Opening Area % | Velocity Vector | Streamline Map |
|--------|---------|---|----------------|-----------------|----------------|
| 70 | 1 | | 0 | A | F |
| 180 | 2 | | 25 | B | G |
| 216 | 3 | | 50 | C | H |
| 290 | 4 | | 75 | D | I |
| | | | 100 | E | J |

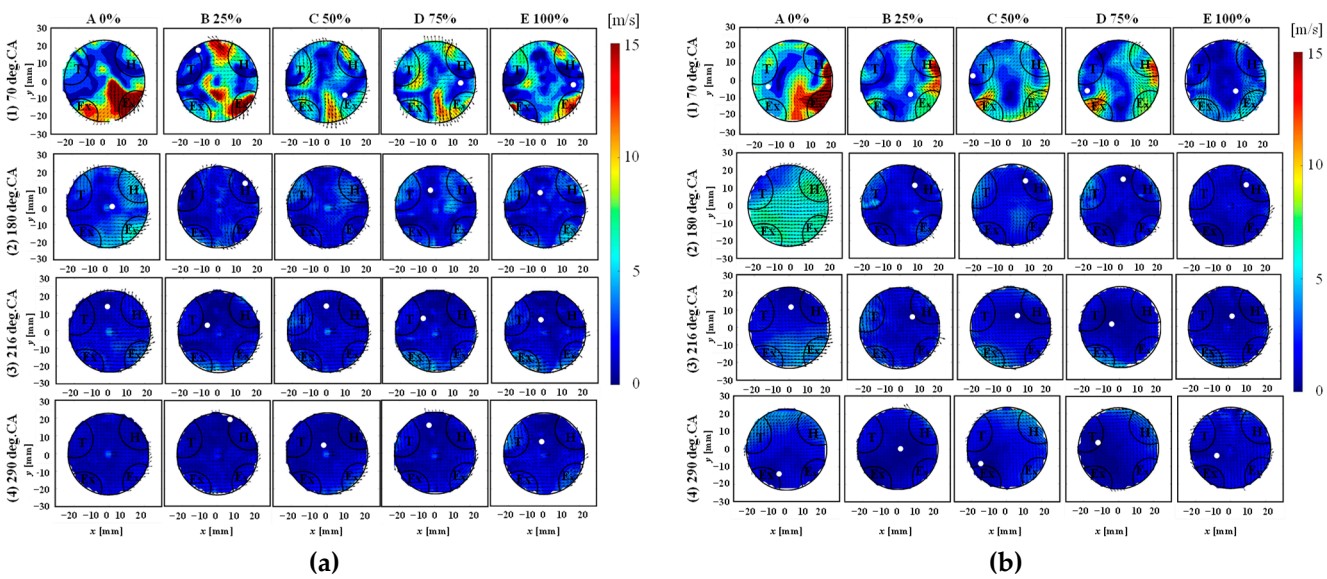

**Figure 3.** Ensemble-averaged velocity vector map at 70 deg.CA, 180 deg.CA, 216 deg.CA, and 290 deg.CA at (**a**) *z* = −10 mm and (**b**) −30 mm under conditions of five different opening areas.

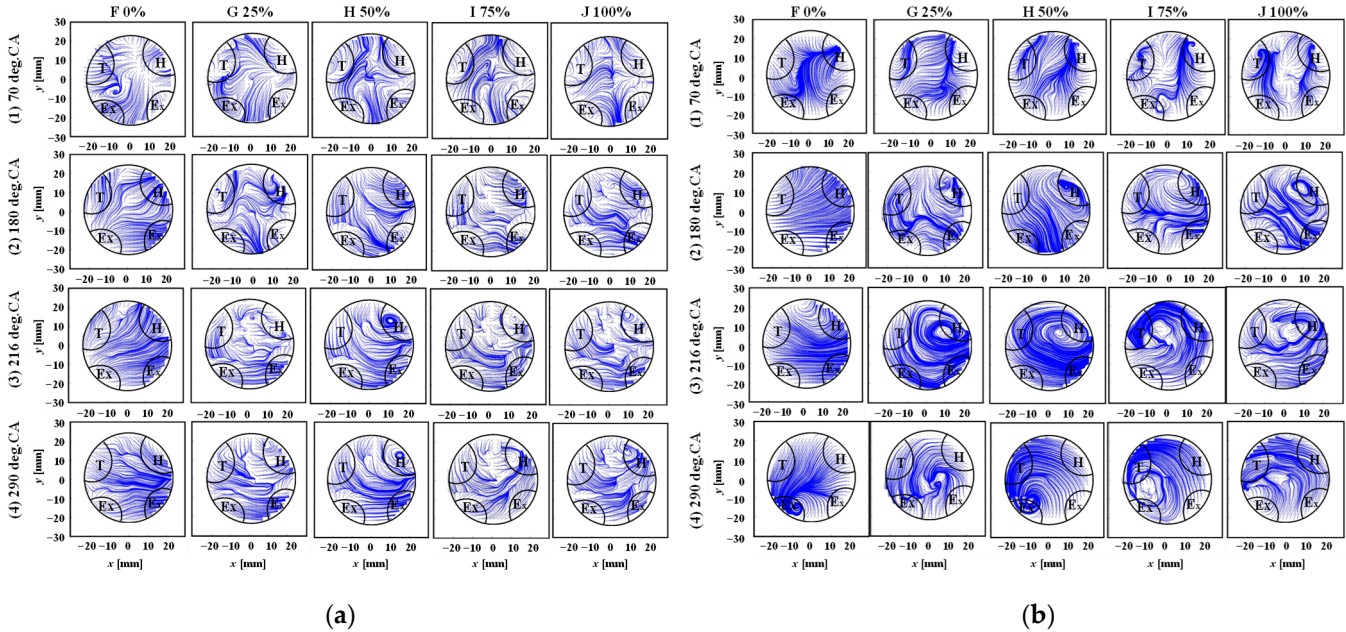

**Figure 4.** Streamline map at 70 deg.CA, 180 deg.CA, 216 deg.CA, and 290 deg.CA at (**a**) *z* = −10 mm and (**b**) −30 mm under conditions of five different opening areas.

### 3.1.1. Intake Stroke

In view of the ensemble-averaged velocity vector maps, at both measurement planes (*z* = −10 mm & −30 mm), it was calculated that the magnitudes of velocities in 1A–1E (during the intake stroke) were larger than those in 2A–4E (during the compression stroke), as shown in Figure 3. In addition, during the intake stroke in each measurement plane, regardless of the port opening areas, streamlines were found to be complicated for each opening area, as shown in Figure 4. This was an indication that the swirl flows were not formed during the intake stroke. In the present study, these types of flows are called "complicated flow".

### 3.1.2. Compression Stroke

From the piston's reciprocating motion, during the compression stroke, the $z = -30$ mm measurement plane shows the earlier results of the in-cylinder flow. Thus, Figure 4b needs to be analyzed prior to Figure 4a. Focusing on the cases with flows only from the helical port (2F–4F), it was observed that the streamlines in 2F–3F had curved lines; however, the SCPs could not be visualized clearly. This was an indication of the continuation of complicated flows. However, as the piston moved upwards, in case 4F, swirl-like flow was observed during the latter period of the compression stroke, as shown in Figure 4b. The SCP for case 4F was calculated as $(x, y) = (-4.6$ mm, $-14.7$ mm$)$. These results make it clear that the initial swirl-like flows were formed towards the end of the compression stroke when the intake air was directed only from the helical port. There was a possibility that a part of the swirl flow (defined as "swirl-like flow") was formed in the cavity, and was caused by the collision of both flows from the tangential and the helical port.

Similarly, at the $z = -30$ mm measurement plane, as the tangential port opening area was increased, the streamlines at the beginning of the compression stroke (in 2G–2J) were still observed to be complicated flow. Consequently, as the piston moved upwards (in 3G–4J) successful formation of swirl flows was observed. In 4G, 4H, 4I, and 4J SCPs were calculated to be $(x, y) = (1.5$ mm, $0.0$ mm$)$, $(-14.9$ mm, $-8.6$ mm$)$, $(-12.0$ mm, $3.4$ mm$)$, and $(0.6$ mm, $-3.6$ mm$)$, respectively. This was apparently the result of the tangential port's effect on the generation of swirl flows from the middle period of the compression stroke. In addition, from our previous studies, we observed that swirl flows were formed at $z = -40$, $-60$, and $-80$ mm under the condition of a 100% opening area for both the helical and tangential ports [15]. Combining the results of the previous and the present study, it can be said that swirl flow was successfully formed 30 mm below the engine cylinder ($z = -30$ mm), with tangential port's opening area of 25% and above, during the compression stroke.

When the focus is moved up to the measurement plane at $z = -10$ mm, in cases 2G–2J, the streamlines were still observed to be complicated flow, similar to those in the $-30$ mm plane, as shown in Figure 4a. However, those in 3G–4J had curved lines, in which SCPs were still visible. For instance, for the 4I and 4J cases, SCPs were calculated as $(x, y) = (-3.1$ mm, $15.4$ mm$)$ and $(-0.8$ mm, $6.5$ mm$)$, respectively. Based on these findings, it is evident that when the tangential port opening areas were 25% or more, as the mixture flow moved upwards inside the engine cylinder, swirl-like flows were maintained. Even though SCPs were not highly apparent, by using both intake ports simultaneously the swirl-like flows continued their effect, which is beneficial for more a homogenous air–fuel mixture to achieve higher combustion efficiency. Table 4 summarizes the classification of flows formed under each condition of the opening areas at each measurement period.

**Table 4.** Classification of flow formed for each opening area of the tangential port and each period at (**a**) $z = -10$ mm and (**b**) $-30$ mm ($\times$: complicated flow, ▲: swirl-like flow, •: swirl flow).

| (a) | | | | | |
|---|---|---|---|---|---|
| **CASE** | **0%** | **25%** | **50%** | **75%** | **100%** |
| 70 deg.CA | $\times$ | $\times$ | $\times$ | $\times$ | $\times$ |
| 180 deg.CA | $\times$ | $\times$ | $\times$ | $\times$ | $\times$ |
| 216 deg.CA | $\times$ | ▲ | ▲ | ▲ | ▲ |
| 290 deg.CA | $\times$ | ▲ | ▲ | ▲ | ▲ |

| (b) | | | | | |
|---|---|---|---|---|---|
| **CASE** | **0%** | **25%** | **50%** | **75%** | **100%** |
| 70 deg.CA | $\times$ | $\times$ | $\times$ | $\times$ | $\times$ |
| 180 deg.CA | $\times$ | $\times$ | $\times$ | $\times$ | $\times$ |
| 216 deg.CA | $\times$ | • | • | • | • |
| 290 deg.CA | ▲ | • | • | • | • |

### 3.1.3. Swirl Ratio

Figure 5a,b illustrates swirl ratio calculated based on the cylinder center, under each tangential opening area at −10 mm and −30 mm, respectively. During the first half of the intake stroke (70 deg.CA—125 deg.CA), results from both measurement planes showed similar tendencies, where a 0% opening area case showed values smaller than zero, meaning an opposite direction of swirl flow generation. However, this swirl flow was diminished rather quickly as the swirl ratio changed to positive values towards the end of first half of the intake stroke. In addition, tangential port opening areas of 25% and above showed a sinusoidal trend, where swirl ratio values were becoming high and low consequently. Thus, in general, it can be said that the formation of swirl flow was not successful during the intake stroke.

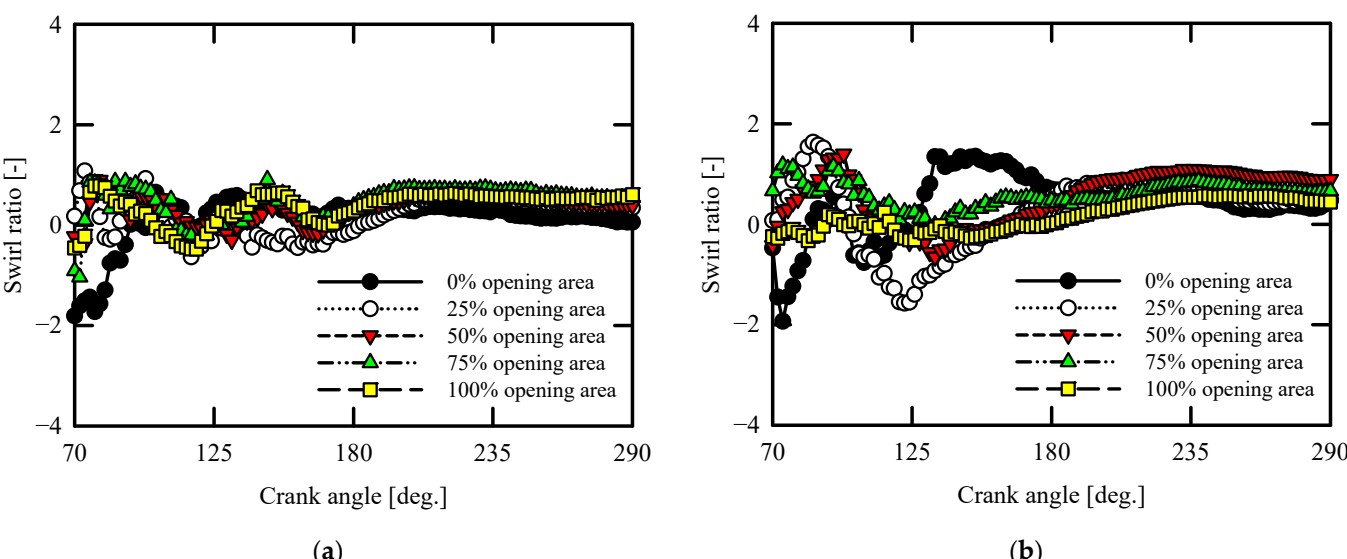

**Figure 5.** Time evolution of swirl ratios calculated from the cylinder center position at (**a**) $z = −10$ mm and (**b**) −30 mm under five different opening area conditions.

During the compression stroke (from 180 deg.CA and onwards), both measurement planes' swirl ratio values showed a somewhat steady level. It also became clear that at $z = −30$ mm the swirl ratio values were higher than those of $z = −10$ mm. This could be interpreted as the in-cylinder air flow velocities in the −10 mm plane becoming slower due to the interference induced by flows returning from the cylinder head. In addition, the sequential order of the swirl ratio values was different at the $z = −10$ mm when compared to $z = −30$ mm plane. At −30 mm, a tangential port opening area of 50% showed the highest value, whereas at −10 mm the highest value was attained under the 100% case. The sequential orders of the swirl ratios for the −10 mm and −30 mm measurement planes were 100% > 75% > 50% > 25% > 0%, and 50% > 75% > 100% > 25% > 0%, respectively. At both measurement planes, for the 0% case (air flow only from the helical port), swirl ratio values showed a decreasing trend. This is thought to be related to the higher in-cylinder velocities attained during the intake stroke from the helical port, causing complicated flow inside the engine cylinder, rather than forming swirl flow. By looking at these results, it became clear that to generate a swirl ratio, the tangential and helical ports need to be used simultaneously.

### 3.2. Evaluation of Spatially Averaged Velocity and Turbulent Kinetic Energy

For the investigation of factors that induced the classification of flows, spatially averaged velocities and TKE were evaluated at $z = −10$ mm and −30 mm, as shown in Figures 6 and 7, respectively. For each case, TKE was calculated using Equation (3). As shown in Figures 6 and 7, during the compression stroke, the spatially averaged velocities and TKE were asymptotic to about 1–2 m/s and about 1–3 $m^2/s^2$, respectively. Although

there were three kinds of flows, as shown in Table 4, large differences in averaged velocities or TKE were not observed during the compression stroke. Thus, it became clear that the intake stroke affected the formation of in-cylinder flows the most, which was also supported by the large variances of TKE during the intake stroke when compared to the compression stroke at various measurement planes.

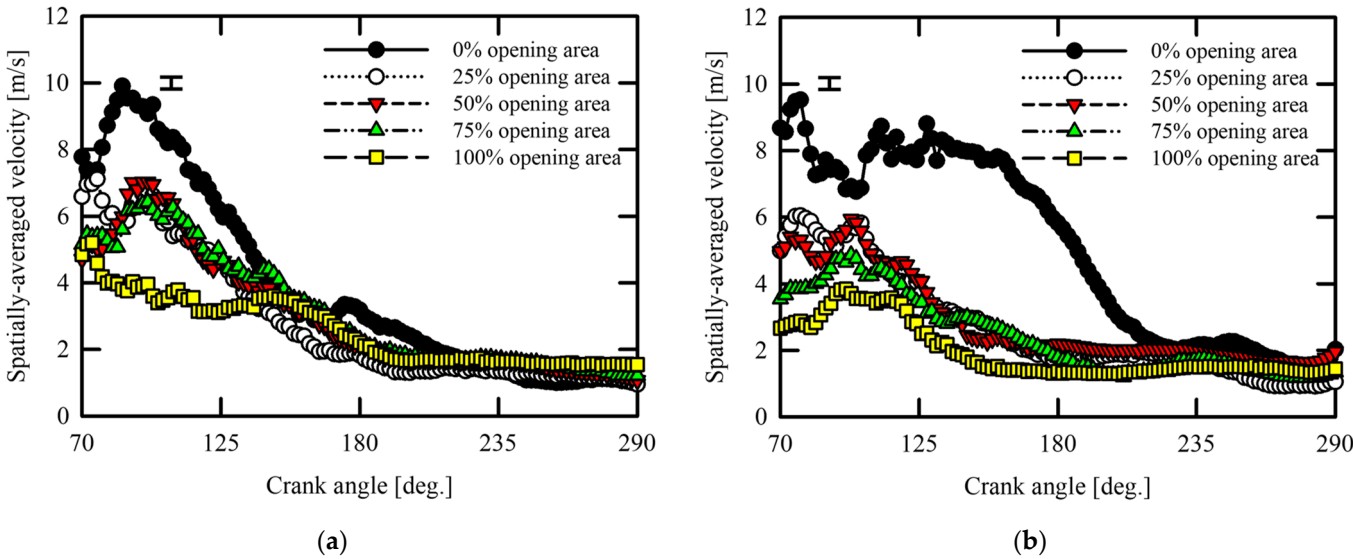

**Figure 6.** Time evolution of spatially averaged velocity at (**a**) $z = -10$ mm and (**b**) $-30$ mm under conditions of five different opening areas. The error bar corresponds to a 95% confidence interval.

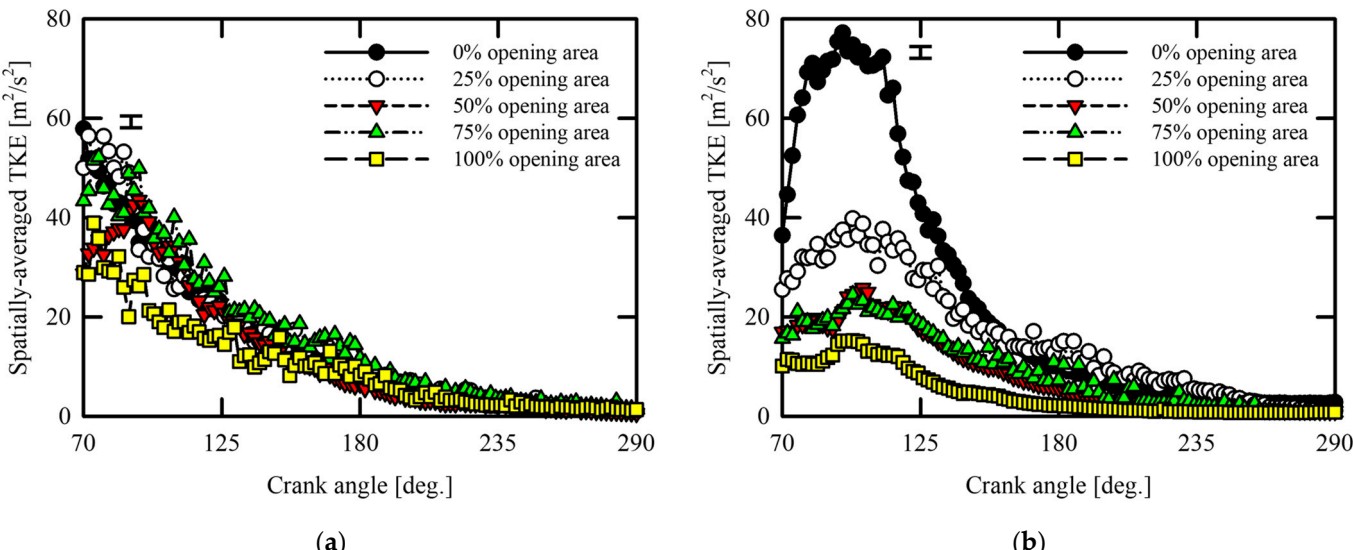

**Figure 7.** Time evolution of spatially averaged TKE at (**a**) $z = -10$ mm and (**b**) $-30$ mm under conditions of five different opening areas. The error bar corresponds to a 95% confidence interval.

At the $z = -10$ mm measurement plane, during the intake stroke (from 70 deg.CA to 180 deg.CA), the order of averaged velocities depending on the tangential port area was calculated to be 0% > 25% ≈ 50% ≈ 75% > 100%, as shown in Figure 6a. Similar results were also apparent for TKE, as shown in Figure 7a, with the exception of the 0% opening area case being similar to 25%, 50%, and 75%. The reasoning behind this was thought to be the similarities between the structure and turbulence of flows; TKE was roughly proportional to the averaged velocity. Thus, orders of both magnitudes were expected to be roughly the same. However, as the opening area of the tangential port was increased, averaged velocities decreased due to the interference between the flows along the cylinder

wall caused by the tangential port and the vertical flow caused by the helical port. In addition, under the condition of a 100% opening area, TKE was smaller than for the 25–75% opening area cases, due to the aforementioned interference flows induced by the opening of both ports. Thus, the order of TKE at the $z = -10$ mm measurement plane was different from the spatially averaged velocities.

For the results at the $z = -30$ mm measurement plane, spatially averaged velocities showed similar results when compared to the $z = -10$ mm measurement plane, with an exception for the 75% opening area case, which briefly showed higher averaged velocities between 125 deg.CA and 180 deg.CA when compared to the 50% opening case. Thus, the order of spatially averaged velocities was 0% > 25% > 50% $\approx$ 75% > 100%, as shown in Figure 6b. The averaged order of TKE was also calculated to be 0% > 25% > 50% $\approx$ 75% > 100%, as shown in Figure 7b. It was also found that under the condition of a 0% opening area, TKE was about twice that of the 25% opening area. This high TKE value for the 0% opening area case is thought to be the reason why there was no formation of swirl flow, as mentioned before (see cases 1F–4F in Figure 4b).

Furthermore, variances of TKE during the intake and compression strokes are listed in Table 5 (a) and (b), respectively. During the intake stroke, at $z = -10$ mm, TKE variances were calculated to be the highest. At this period, there was no formation of swirl flow under all tangential port opening areas. Similarly, at $z = -30$ mm under the condition of a 0% opening area, TKE variance was calculated to be high, with no formation of swirl flow. Conversely, when TKE variances became smaller during the compression stroke (at $z = -30$ mm), swirl flows were formed, as already shown in Figure 4b. From these results, it became clear that almost no swirl flow was formed when the TKE variances were large. Meaning that, in order to generate swirl flow, TKE variances needs to be small, which was achieved by using both the tangential and helical ports at the same time.

**Table 5.** Variances of TKE averaged during (**a**) the intake and (**b**) the compression strokes under conditions of five different opening areas. The unit is $m^4/s^4$.

| (a) | | | | | |
|---|---|---|---|---|---|
| **CASE** | **0%** | **25%** | **50%** | **75%** | **100%** |
| $z = -10$ mm | 180.0 | 219.9 | 136.3 | 151.7 | 59.9 |
| $z = -30$ mm | 563.0 | 73.6 | 36.3 | 24.7 | 18.3 |
| (b) | | | | | |
| **CASE** | **0%** | **25%** | **50%** | **75%** | **100%** |
| $z = -10$ mm | 4.7 | 4.4 | 2.2 | 6.2 | 3.7 |
| $z = -30$ mm | 4.5 | 15.1 | 1.4 | 5.3 | 0.2 |

*3.3. Evaluation of Swirl Center Positions*

For further investigation of the classification of flows, SCPs at $z = -10$ mm, $-20$ mm, and $-30$ mm under the conditions of five different opening areas were evaluated using the technique described in Section 2.3. Figure 8a shows the SCP maps in the $x$–$y$ plane during the intake stroke between 70 deg.CA and 180 deg.CA, whereas Figure 8b shows the latter period of the compression stroke between 254 deg.CA and 290 deg.CA

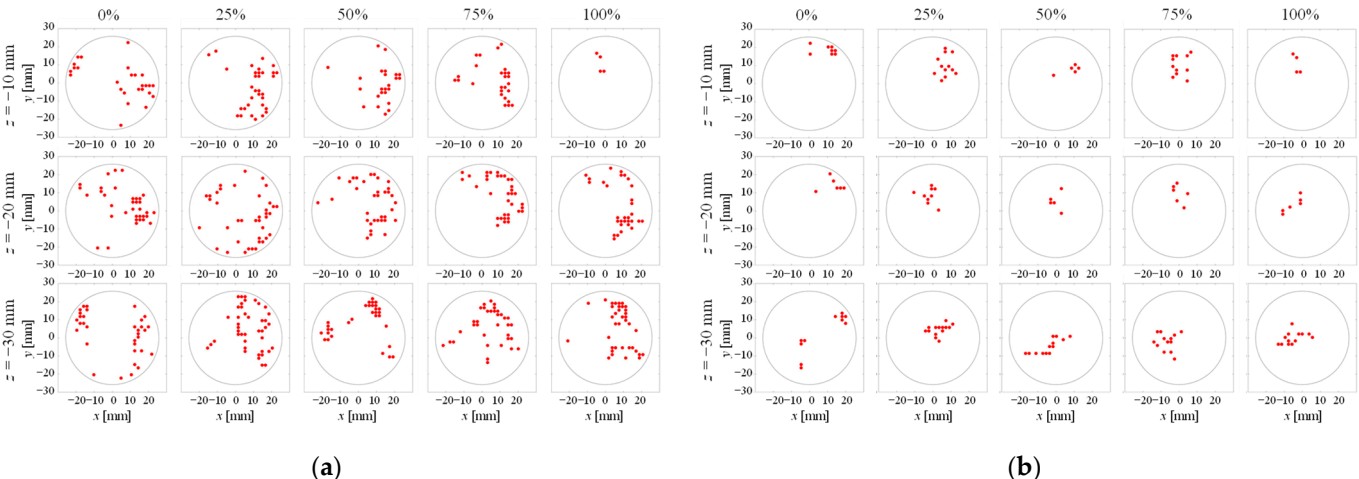

(**a**)                                                    (**b**)

**Figure 8.** (**a**) SCP maps during the intake stroke and (**b**) the latter period of the compression stroke in the *x*–*y* plane under conditions of five different opening areas.

During the intake stroke, it was observed that SCPs tend to be dispersed regardless of the measurement plane and the opening area. Since these results correspond to 1F–2J of Figure 4b, it was considered that these dispersions of SCPs were induced by the formation of complicated flow as TKE variances were also high. Thus, no clear correlation could be made for the formation of swirl flow during the intake stroke.

During the latter period of the compression stroke, it was more apparent that SCPs were concentrated within a certain region at all measurement planes. However, SCPs differed depending on opening area conditions of the tangential port, as shown in Figure 8b. Similar to the results given in cases 4F–4J of Figure 4a,b, corresponding to $z = -10$ mm and $-30$ mm, respectively, swirl like flows were observed at $z = -10$ mm, whereas swirl flows were formed at $-30$ mm during the latter period of the compression stroke.

From the viewpoint of the quantitative evaluation, the variances of SCPs were calculated at $z = -10$ mm and $-30$ mm. Intake stroke results are given in Table 6 (a), and the latter period of the compression stroke results are given in Table 6 (b). During the intake stroke, SCP variances were larger than those during the latter period of the compression stroke under the conditions of 25–100% opening areas. This was due to the higher in-flow velocities induced during the intake stroke, causing high variances for the exact location of SCPs for each case.

**Table 6.** Variances of SCPs averaged during (**a**) the intake stroke and (**b**) the latter period of the compression stroke under conditions of five different opening areas. The units are mm$^2$.

| (a) | | | | | |
|---|---|---|---|---|---|
| **CASE** | **0%** | **25%** | **50%** | **75%** | **100%** |
| $z = -10$ mm | 180.0 | 219.9 | 136.3 | 151.7 | 59.9 |
| $z = -30$ mm | 563.0 | 73.6 | 36.3 | 24.7 | 18.3 |
| (b) | | | | | |
| **CASE** | **0%** | **25%** | **50%** | **75%** | **100%** |
| $z = -10$ mm | 4.7 | 4.4 | 2.2 | 6.2 | 3.7 |
| $z = -30$ mm | 4.5 | 15.1 | 1.4 | 5.3 | 0.2 |

For the latter period of the compression stroke, Table 6 (b), SCP variances were lowered as these flows corresponded to swirl-like or swirl flows. However, SCPs changed depending on the measurement plane and the tangential port's opening area. Thus, from a qualitative evaluation perspective, the relationship between the SCPs and the flow structure is unclear.

In order to have a clearer idea, the proper orthogonal decomposition (POD) and MODE calculations will be implemented to further understand the in-cylinder flow structure.

Figure 9 shows the SCP locations averaged during the latter period of the compression stroke, at (a) the *x–z* plane and (b) the *y–z* plane. It should be noted that these values are averaged results, in which some cases of SCP locations differed due to in-cylinder air flow. As already shown in Figure 1b, both intake ports were located at the positive region in the y direction ($x > 0$ and $y > 0$ for the helical port, and $x < 0$ and $y > 0$ for the tangential port). Thus, it is considered that the in-cylinder flow was formed with the tilt of the SCP, due to the locations of the intake ports and the interference between the flows induced from them depending on the piston's position in the *z*-axis. In Figure 9a, for the 0% opening area case, it was observed that SCPs were centered towards the positive *x* direction at all measurement planes, where the helical port was located. As the tangential port's opening area was increased, with the piston's upward movement (from −30 mm to −10 mm), SCPs were shifted towards the center. For the analyses in the *y* direction (Figure 9b), since both intake ports were located in a positive *y* direction, under all opening areas, SCPs were located towards the positive region of the *y* direction.

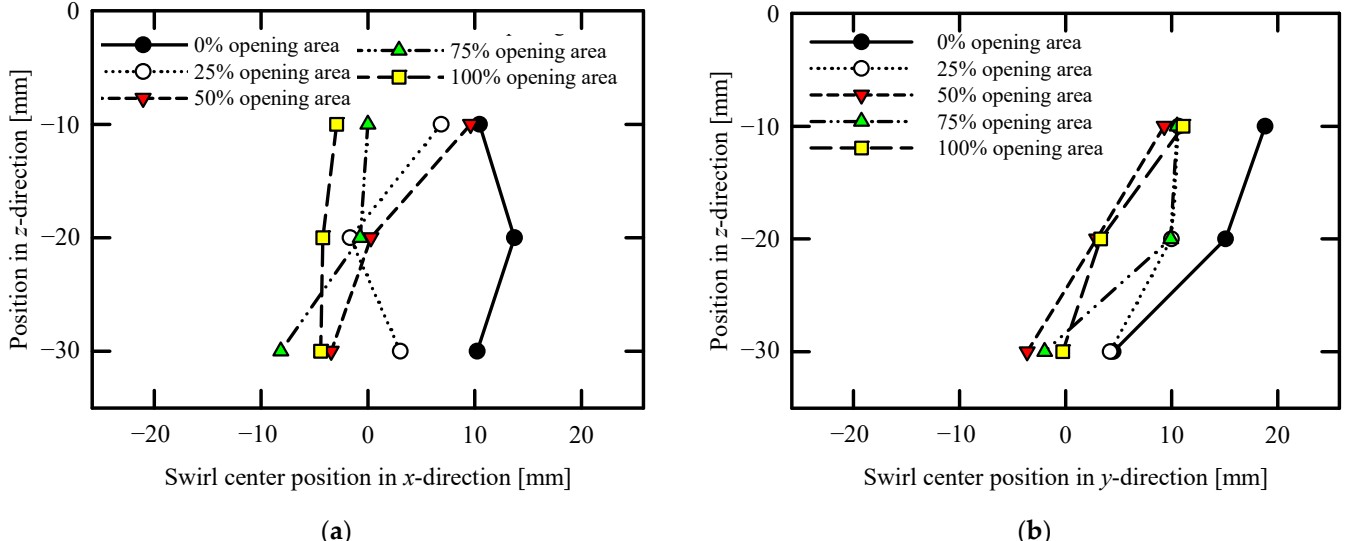

**(a)**　　　　　　　　　　　　　　　　**(b)**

**Figure 9.** (**a**) SCPs averaged from 254 deg.CA to 290 deg.CA (latter period) in the *x–z* plane and (**b**) the *y–z* plane under conditions of five different opening areas.

Summarizing the above, in an engine when both tangential and helical ports were used, it was expected to observe complicated or swirl-like flows when the SCP variances were relatively large. In contrast, swirl flows were formed when SCP variances were relatively small, which was induced by the higher opening areas of the tangential port.

## 4. Conclusions

For a four-valve diesel engine, the effects of the opening area of the tangential port on the in-cylinder swirl flow were investigated, where the opening areas were changed to 0%, 25%, 50%, 75%, and 100% using different gaskets. The present study measured the velocities in a transparent engine cylinder in three different measurement planes, at $z = -10$ mm, −20 mm, and −30 mm, by using the PIV technique. The gathered experimental data were used to evaluate velocity vectors, streamline maps, spatially averaged TKE, and SCPs in three-dimensional coordinates. Important conclusions obtained from the present study are summarized below:

1.  The velocity vector and the streamline maps were evaluated using the obtained velocities through PIV measurements. In the case of the 0% opening area, during intake stroke, complicated flows were observed at $z = -10$ mm, −20 mm and −30 mm, where variances of TKE and SCPs were large. During the compression stroke, com-

plicated flows were also observed at $z = -10$ mm and $-20$ mm measurement planes. At the latter period of the compression stroke, swirl-like flows started to form at $z = -30$ mm.

2. In the case of opening areas of 25% or more (flows from both ports), during intake stroke, similar tendencies were observed in the cases where only the helical port was used, where no swirl flow generation was observed. After the first half of the compression stroke, swirl-like flows were observed at $z = -10$ and $-20$ mm and swirl flows were successfully formed at $z = -30$ mm.

3. Swirl ratios were calculated from the center of the engine cylinder for each tangential port opening area. Calculations showed that, during the intake stroke, a sinusoidal pattern was apparent in the swirl ratios, meaning proper swirl flow was not formed. On the other hand, during the compression stroke, as the tangential port opening areas increased, swirl ratio also increased and reached to a steady level for cases when the tangential port opening was 25% and above.

4. The spatially averaged TKE and its variances were evaluated using the obtained velocities. During the compression stroke, large differences in TKE were not observed, thus a comparison cannot be made. However, during the intake stroke differences in the TKE were apparent depending on the measurement planes and the size of the opening areas. It was concluded that, during the intake stroke, as the variances of TKE became larger, complicated or swirl-like flows were formed. As these variances started to become smaller, swirl flows started to form.

5. SCPs and their variances were evaluated using the obtained velocities. SCPs were not clear during the intake stroke. However, for cases with an opening area of 25% or more, SCPs were observed clearly during the compression stroke. A tilting motion of SCPs was also observed in the $x$–$y$ planes in the $z$ direction during the compression stroke. The SCP variances under the conditions that form complicated or swirl-like flows were larger than those under the conditions that form swirl flows. It was concluded that swirl flows were successfully formed when the variances of SCPs were relatively low.

As the next step of this study, the know-how on the swirl flow formation will be applied to a modified diesel engine with a high compression ratio, which is fueled by an ammonia–gasoline mixture. In this setup, the fuel mixture of ammonia–gasoline will be used in a port injection method. Subsequent experiments will be conducted to validate the outcomes presented in this paper through ammonia–gasoline co-firing.

**Author Contributions:** Conceptualization, M.I. and T.S.; data curation, M.I., K.H. and T.H.; formal analysis, M.I.; funding acquisition, M.I. and T.S.; investigation, M.I., E.Y., K.H., T.H., W.A. and T.S.; methodology, M.I. and T.S.; project administration, M.I. and T.S.; resources, M.I. and T.S.; software, K.H. and T.H.; supervision, M.I. and T.S.; validation, M.I., E.Y. and T.S.; visualization, M.I., E.Y. and T.S.; writing—original draft, M.I., E.Y., K.H. and T.H.; writing—review and editing, M.I., E.Y., W.A. and T.S. All authors have read and agreed to the published version of the manuscript.

**Funding:** This research was funded by the Japan Society for the Promotion of Science, Grants-in-Aid for Scientific Research (No. 19K04244).

**Data Availability Statement:** Data will be available from the corresponding author upon reasonable request. All other data supporting this study are available within the article.

**Acknowledgments:** The authors thank Yoseph A. Salim and Shunya Aoyagi for their technical support.

**Conflicts of Interest:** The authors declare no conflict of interest.

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
