# Peer review of "Experimental Investigation of the In-Cylinder Flow of a Compression Ignition Optical Engine for Different Tangential Port Opening Areas"

_energies, doi:10.3390/en16248110_

Round 1

Reviewer 1 Report

Comments and Suggestions for Authors

Opinion 1

line224-226

In the view of ensemble-averaged velocity-vector maps, at both measurement planes (z = -10 mm & - 30 mm), it was calculated that the magnitudes of velocities in 1A-1E are larger than those in 2A-4E (during compression stroke), as shown in Figures 3 (a) and 4 (a).

Opinion:The measurement plane (z=-30mm) was not elaborated in the Figures 3 (a) and 4 (a).

Opinion 2

line232-235

In Figure 4 (b) Starting from case 2F to 4F (whole compression with flows from helical 232 port only), it was observed that the streamlines in 2F-3F had curved lines, however, SCPs could not be visualized clearly. This was an indication of generation of complicated flow, similar to the intake stroke case.

Opinion:CASE 2F-3F can just show that a complex flow is generated, not that it is the same as the intake stroke case. There is an ambiguity in the expression.

Opinion 3

Line290-292

However, during the intake stroke at z = -30 mm, the order of averaged velocities and TKEs were more comparable. The order of spatially-averaged velocities was 0% > 25% > 50% > 75% > 100%, as shown in Figure 5 (b).

Opinion:At CA125-CA180 on the intake stroke, the space-averaged velocity briefly appears to be 75% greater than 50%, which is inconsistent with the author's conclusions.

Opinion 4

Line359-361

For the analyses in the y-direction (Figure 8 (b)), since both intake ports were located in positive y-direction, under all opening areas SCPs were located towards the positive region of the y-direction.

Opinion:The opening areas are 0% and Z = -30 mm, the scp position in Fig. 7b has a significant appearance in the y-negative half-axis, which is inconsistent with the conclusions given by the author.

Comments on the Quality of English Language

No comments.

Author Response

We have included the responses to the comments in the attached PDF.

Reviewer 2 Report

Comments and Suggestions for Authors

The following suggestions/questions can be considered by the authors to improve the quality of the present manuscript.

1. “The push for internal combustion engine (ICE) decarbonization has spurred interest in alternative fuels like hydrogen and ammonia. Due to their unique properties, such as viscosity and density, it is necessary to further focus on the intake system and in-cylinder flows to enhance combustion efficiency and lower emissions.” So, which kind of fuel is used in research? Hydrogen or ammonia? Please provide more information.

2. In Figure 2, the shape and orientation of helical and tangential intake ports are fixed, whether consideration will affect the generation of swirl flows, please add more information.

3. In the Introduction, we know that swirl ratio is vital to ICE performance, and “We found that by changing the helical port’s opening areas, SCPs do not shift significantly, and airflow velocities do not change with an increase in engine speed. In addition, as the helical port’s opening area was increased, the swirl ratio was reduced, which lowered the turbulent intensity”, now our attention shifted to the tangential port instead, how about the swirl ratio, it seems not to be mentioned, please provide more materials and information.

Author Response

(The authors gave the same response as above.)

Reviewer 3 Report

Comments and Suggestions for Authors

Thank you for this interesting submission presenting velocimetry results for an optically-accessible CI engine. The experimental setup is well-described and the results are thoroughly documented. I do have a few questions and suggestions for the paper:

- You may want to consider a slight change to the title. I think adding the word "Areas" at the end could make the meaning clearer. This is optional, I leave it to your discretion.

- At Line 94, you say "there are only a few studies where tangential port's effect was experimentally investigated". Which studies? What was learned, and what, if anything, is different in your study?

- Notes should be added to Figure 1b and 2 specifying that dimensions are in millimeters.

- In Figure 2, it is difficult to discern what all the lines inside the two ports are meant to represent. Consider simplifying the figure.

- I was somewhat confused about Section 3.1.2. The first paragraph discusses Figure 4 (b) at the z = -30 mm location. However, it suggests that cases 4I and 4J have swirl-like flow at z = -30 mm, while Table 4 says that those cases at z = -30 mm have swirl flow. Then the next paragraph begins with "on the other hand, at z = -30 mm" as if the first paragraph concerned a different location. I suspect that the first paragraph is actually meant to describe Figure 4 (a) at z = -10 mm. Please review and clarify.

- You describe "variations of SCP" in and around Table 6. I think you mean "variances", i.e. the square of the standard deviation.

- You began your paper describing why it was important to know how tangential port opening area affects airflow: you want to understand the effects on fuel-air mixing in order to increase performance of alternative fuels. The results and conclusions are a very thorough report of what you found, but there is no discussion about what those results mean for real engines. So, how do you expect tangential port opening area to affect fuel-air mixing or performance? Or, if this question cannot yet be answered, what are the next steps?

- Your introduction mentions CFD analyses of the effects of tangential port opening area in Ref. 19 as well as some experimental studies. How do your results compare?

Author Response

(The authors gave the same response as above.)

Round 2

Reviewer 1 Report

Comments and Suggestions for Authors

No comments for authors.

Comments on the Quality of English Language

No comments on the quality of English language.